# Peripheral Blood Biomarkers Reveal Dysregulated Monoaminergic Pathways in Obsessive–Compulsive Disorder: A Transcriptional and Epigenetic Analysis

**DOI:** 10.3390/ijms26188811

**Published:** 2025-09-10

**Authors:** Fabio Bellia, Nicolaja Girone, Beatrice Benatti, Matteo Vismara, Mauro Pettorruso, Giovanni Martinotti, Bernardo Dell’Osso, Claudio D’Addario, Mariangela Pucci

**Affiliations:** 1Department of Bioscience and Technology for Food, Agriculture and Environment, University of Teramo, 64100 Teramo, Italy; fabio.bellia@unich.it; 2Center for Advanced Studies and Technology (CAST), “G. D’Annunzio” University of Chieti-Pescara, 66100 Chieti, Italy; 3Department of Psychiatry, Department of Biomedical and Clinical Sciences “Luigi Sacco”, University of Milan, ASST Fatebenefratelli-Sacco, 20157 Milan, Italy; girone.nicolaja@asst-fbf-sacco.it (N.G.); beatrice.benatti@unimi.it (B.B.); matteo.vismara@unimi.it (M.V.); bernardo.dellosso@unimi.it (B.D.); 4“Aldo Ravelli” Center for Nanotechnology and Neurostimulation, University of Milan, 20157 Milan, Italy; 5Department of Neuroscience and Imaging, “G. D’Annunzio” University of Chieti-Pescara, 66100 Chieti, Italy; mauro.pettorruso@unich.it (M.P.); giovanni.martinotti@unich.it (G.M.); 6Bipolar Disorders Clinic in the Department of Psychiatry and Behavioral Sciences, Stanford University, Stanford, CA 94305, USA; 7Department of Clinical Neuroscience, Karolinska Institutet, 17177 Stockholm, Sweden

**Keywords:** neuropsychiatric, obsessive–compulsive disorder, peripheral blood mononuclear cells, monoaminergic neurotransmitter, epigenetics

## Abstract

This study investigated the complexity of neurotransmitter-related gene regulation in peripheral blood mononuclear cells (PBMCs) of patients with obsessive–compulsive disorder (OCD), aiming to identify clinically relevant molecular markers. We analyzed three key genes: *SLC6A4* (serotonin transporter), *MAOB* (monoamine oxidase B, a dopamine-degrading enzyme), and *COMT* (catechol-O-methyltransferase, a dopamine/norepinephrine metabolizing enzyme). OCD patients exhibited significant downregulation of *SLC6A4* and *MAOB*, accompanied by upregulation of *MB-COMT*. The contrasting expression of *MAOB* and *MB-COMT* suggests a dysregulated compensatory mechanism in dopamine homeostasis, which contributes to clinical heterogeneity and variability in treatment for OCD. Epigenetic analysis revealed that downregulation of *SLC6A4* was associated with hypermethylation of the gene promoter, demonstrating the critical role of epigenetic mechanisms in neurotransmitter system dysregulation. Moreover, gene–gene correlations identified distinctive molecular expression patterns that reliably discriminated OCD patients from healthy individuals, proposing their potential as peripheral biomarkers. In conclusion, serotonergic and dopaminergic abnormalities characterize OCD, where epigenetic regulation contributes to gene dysregulation. The identified molecular signatures may explain the inefficiency of treatments and support biomarker-guided clinical approaches. Given that peripheral gene regulation and core neurotransmitter systems are similar, this study contributes to the biological picture of OCD, indicating the accuracy of diagnoses and treatments.

## 1. Introduction

Obsessive–compulsive disorder (OCD) is a debilitating neuropsychiatric condition characterized by recurrent, intrusive thoughts (obsessions) and repetitive behaviors or mental acts (compulsions) that significantly impair daily functioning. With a lifetime prevalence of approximately 2–3% worldwide, OCD ranks among the leading causes of disability. However, the molecular mechanisms underlying this disorder remain incompletely understood [1,2,3].

The heterogeneous nature of OCD symptomatology, coupled with variable treatment responses, suggests that complex genetic and epigenetic factors contribute to disease pathophysiology [4]. Accumulating evidence supports the involvement of monoaminergic neurotransmitter systems, particularly serotonin and dopamine pathways, in OCD pathogenesis [5,6,7].

Central to these systems are key regulatory proteins that modulate neurotransmitter availability and metabolism. The serotonin transporter (SERT), encoded by the *SLC6A4* gene, represents the primary mechanism for serotonin reuptake from synaptic clefts and has been extensively implicated in OCD through both genetic association studies and pharmacological evidence from selective serotonin reuptake inhibitors (SSRIs) [8,9]. While SSRIs are typically the first-line therapeutic approach and can significantly reduce symptoms in many individuals, this therapy proves ineffective in 40–60% of patients [10,11]. For non-responding patients, antipsychotics are often prescribed alongside cognitive behavioral therapy (CBT) [12,13].

Monoamine oxidase B (MAOB), a mitochondrial enzyme responsible for dopamine and phenylethylamine catabolism, has emerged as a critical regulator of dopaminergic signaling, which is increasingly recognized as dysregulated in OCD [14,15]. Additionally, membrane-bound catechol-O-methyltransferase (MB-COMT), distinct from its soluble cytoplasmic counterpart, plays a specialized role in dopamine and norepinephrine metabolism at synaptic terminals, potentially contributing to the dopaminergic abnormalities observed in OCD patients [16,17].

While neuroimaging and postmortem brain tissue studies have provided valuable insights into OCD neurobiology [18,19,20,21], accessibility limitations have necessitated the exploration of peripheral biomarkers that may reflect the central nervous system pathophysiology. Peripheral blood mononuclear cells (PBMCs) have emerged as a promising model for investigating neuropsychiatric disorders due to their shared developmental origin with neural tissues and responsiveness to similar regulatory mechanisms [22,23]. Importantly, PBMCs express neurotransmitter receptors, transporters, and metabolic enzymes, making them suitable surrogates for studying gene expression patterns that may parallel those in the central nervous system [24]. We have previously demonstrated that altered expression of specific genes observed in human peripheral tissues reflects altered expression of the same genes in the CNS of OCD-like preclinical models [25,26], emphasizing the importance of identifying peripheral biomarkers to track disease presence and progression.

Complex regulatory networks involving transcriptional, post-transcriptional, and epigenetic mechanisms are crucial for gene expression regulation. DNA methylation, microRNA (miRNA), and histone modifications contribute to dynamic gene expression control in response to environmental and pathological stimuli. Understanding these regulatory mechanisms in OCD may provide valuable insights into the disease’s underlying biology and help identify potential therapeutic targets.

This study investigates the regulatory mechanisms of critical neurotransmitter-related genes in PBMCs from OCD patients. By examining the expression levels of *SLC6A4*, *MAOB*, and *MB-COMT*, along with their associated regulatory elements, we aim to identify peripheral molecular signatures that reflect the dysregulated monoaminergic signaling characteristic of OCD. Such findings could facilitate the development of objective biomarkers for OCD diagnosing, prognosis, and treatment monitoring while enhancing our fundamental understanding of genetic and epigenetic factors involved in obsessive–compulsive symptoms.

## 2. Results

### 2.1. Gene Expression Analysis

The primary finding of this study was a significant variation in mRNA abundance between OCD patients and healthy controls across all three target genes. We observed significant downregulation of *SLC6A4* (CTRL: 1.067 ± 0.079; OCD: 0.185 ± 0.025, *p* < 0.0001) (Figure 1b) and *MAOB* (CTRL: 1.236 ± 0.158; OCD: 0.138 ± 0.019, *p* < 0.0001) (Figure 2b).

In contrast, *MB-COMT* showed significant overexpression (CTRL: 1.118 ± 0.098; OCD: 2.450 ± 0.263, *p* < 0.0001) (Figure 3b) in OCD subjects compared to healthy controls.

Sex-stratified analysis revealed consistent patterns across both male and female participants, with no sex-specific trends for any of the three genes examined (Appendix A). Detailed sex-stratified results are presented in Appendix A.

### 2.2. DNA Methylation Analysis

We analyzed DNA methylation levels in CpG islands within the promoter region of the studied genes. For the *SLC6A4* promoter region, we examined 6 CpG sites (Figure 1a) and found that DNA methylation levels were significantly increased in OCD subjects compared to controls. At CpG site 2, the difference was statistically significant, with OCD subjects showing higher methylation levels than healthy individuals (CTRL: 4.089 ± 0.127; OCD: 4.560 ± 0.154, *p* = 0.0024) (Figure 1c).

However, we observed no significant differences in DNA methylation levels at the 5 CpG sites in the *MAOB* promoter region (Figure 2c) or the 8 CpG sites in the *COMT* promoter region (Figure 3c). Sex-stratified analysis showed only a tendency for increased *SLC6A4* DNA methylation in both male and female OCD patients compared to controls, but no statistically significant differences were found (Appendix A). Detailed individual CpG site results are provided in Appendix A.

### 2.3. Correlation Analysis

We examined the relationship between relative gene expression (2^−ΔΔCt^ values) and DNA methylation percentages at *SLC6A4* CpG site 2, which differed between groups. Our analysis revealed a significant negative correlation for both OCD patients and healthy subjects (Spearman’s r = −0.3554, *p* = 0.0459) (Figure 1d).

Investigation of intergenic interactions revealed strong correlations among the three target genes. Specifically, *SLC6A4* and *MAOB* expressions showed an extremely high positive correlation (Spearman’s r = 0.8318, *p* < 0.0001). Conversely, we observed a strong negative correlation between *SLC6A4* and *MB-COMT* (Spearman’s r = −0.4602, *p* = 0.0009) and between *MB-COMT* and *MAOB* (Spearman’s r = −0.4177, *p* = 0.0028) (Figure 4).

Notably, examination of subject distribution in the correlation graphs revealed a distinct separation between the OCD and healthy control groups based on the three-gene interaction pattern (see Figure 5). The individual group correlation statistics are detailed in Appendix A.

## 3. Discussion

In this study, we analyzed the regulation of genes involved in serotonin and dopamine transport and metabolism using nucleic acids from PBMCs of OCD patients and healthy controls. Our primary observation was a significant downregulation of *SLC6A4* and *MAOB* genes in OCD patients compared to healthy controls. The *SLC6A4* gene encodes the serotonin transporter (SERT/5-HTT), while *MAOB* encodes the enzyme responsible for dopamine degradation in the brain, primarily in astrocytes and radial glia [27].

The selection of these targets was based on the established roles of dopamine and serotonin as crucial neurotransmitters regulating mood, movement, reward, and other functions [28,29,30]. While the synthesis, release, and recycling of these neurotransmitters differ between brain and peripheral tissues, such as immune cells [31,32], PBMCs may mirror the CNS gene regulation status [33,34,35]. Previous investigations by our group and others have reported the selective modulation of neurotransmitter system-related genes in PBMCs across various psychiatric and neurological disorders [26,36,37,38].

### 3.1. Gene-Specific Findings in the Context of Existing Literature

Both serotonin and monoamine oxidase are implicated in OCD, though their relationship remains incompletely understood. Serotonin plays a crucial role in OCD, and SSRIs remain the most effective pharmacological treatment [13,39,40], despite ineffectiveness in a substantial proportion of patients [10,11].

Previous studies investigating peripheral *SLC6A4* mRNA in OCD have yielded contradictory results. Some found no significant differences compared to healthy subjects [41,42], while one reported *SLC6A4* upregulation in OCD subjects compared to controls; however, this group had a comorbidity of Gilles de la Tourette syndrome (GTS) [43]. These inconsistencies may reflect methodological differences, sample heterogeneity, or the influence of comorbid conditions, highlighting the importance of our more focused approach.

Complementing our serotonin transporter findings, we observed significant *MAOB* downregulation in OCD patients. MAO enzymes, which metabolize serotonin, also appear to be relevant, with studies suggesting links between MAO activity and OCD [44,45,46,47,48]. However, no studies have previously examined peripheral *MAO* expression in OCD. Kandemir and colleagues found upregulation of miR22-3p, a microRNA regulating *BDNF* and *MAOA* [49], in pediatric OCD patients, suggesting post-transcriptional *MAOA* downregulation [50]. Our direct measurement of *MAOB* expression provides novel evidence for monoamine oxidase dysregulation in OCD.

In contrast to the downregulation observed for *SLC6A4* and *MAOB*, an increased expression of *MB-COMT* was detected in OCD patients in comparison to the control group. This gene encodes the enzyme responsible for dopamine and norepinephrine degradation. A specific *COMT* gene variation (Val158Met–rs4680) has been studied in relation to OCD, with some literature suggesting associations between this polymorphism and the disorder, particularly in males [48,51,52,53,54,55,56]. Only one previous study investigated *COMT* expression in OCD, observing downregulation in patients compared to controls, with more pronounced effects in women [57]. Our contrasting finding of upregulation may reflect differences in study populations, methodological approaches, or the specific *COMT* isoform examined.

A particularly striking finding was the opposite regulation of the two key dopaminergic enzymes: While *MAOB* was significantly downregulated, *MB-COMT* showed marked upregulation in OCD patients. This opposing pattern suggests compensatory mechanisms within the dopaminergic system that may reflect the brain’s attempt to maintain dopamine homeostasis under pathological conditions.

From a theoretical perspective, the downregulation of *MAOB* would lead to reduced dopamine catabolism, potentially increasing dopamine availability in glial compartments. However, the simultaneous upregulation of *MB-COMT* would enhance dopamine degradation at synaptic terminals, creating a complex regulatory scenario. This antagonistic enzyme regulation may represent a dysregulated compensatory mechanism where the system attempts to balance dopamine levels but fails to achieve proper homeostasis.

The spatial organization of these enzymes provides additional insight into this regulatory pattern. MAOB primarily metabolizes dopamine in astrocytes and glial cells, while MB-COMT acts at synaptic terminals. Their opposing regulation might suggest that different cellular compartments may be attempting to compensate for dopaminergic dysfunction through distinct mechanisms. This may create spatially heterogeneous dopamine availability that could contribute to the complex symptomatology observed in OCD.

### 3.2. Epigenetic Mechanisms Underlying Gene Expression Changes

In view of the gene expression results, an investigation was conducted into the potential epigenetic mechanisms that modulate differentially expressed genes. We focused on DNA methylation at selective CpG sites within the promoter region CpG islands of the *SLC6A4*, *MAOB* and *COMT* genes. We observed significant differences in DNA methylation specifically at the *SLC6A4* promoter when comparing OCD patients to healthy controls. OCD patients showed higher DNA methylation at CpG site 2. Increased DNA methylation is associated with reduced DNA accessibility to the transcription machinery [58], which corresponds to decreased gene expression in these individuals.

Furthermore, we found a significant negative correlation between mRNA and DNA methylation levels at CpG site 2, supporting *SLC6A4* regulation by promoter region DNA methylation. Our methylation findings align with emerging evidence for epigenetic dysregulation in OCD. Previous research has examined *SLC6A4* promoter DNA methylation in OCD, with one study reporting increased methylation in the saliva of pediatric OCD patients compared to controls and adult patients [42]. Other studies have suggested that baseline DNA hypomethylation at the *SLC6A4* promoter in OCD patients could predict impaired treatment response following 10 weeks of cognitive behavioral therapy [59]. The demonstrated role of DNA methylation in regulating the *SLC6A4* gene suggests the potential for epigenetic therapies. Demethylating agents or histone deacetylase inhibitors could be explored as adjunctive treatments, particularly in patients showing hypermethylation patterns associated with treatment resistance.

### 3.3. Gene Correlation Patterns and Population Stratification

Our correlation analysis revealed additional layers of complexity in the regulatory relationships between these neurotransmitter system genes. The sex-stratified analysis revealed no sex-biased transcriptional regulation for any of the examined genes. However, our relatively small sample size may limit findings generalizability. Notably, we identified a positive correlation between *SLC6A4* and *MAOB*, as well as significant negative correlations between *SLC6A4* and *MB-COMT*, and between *MAOB* and *MB-COMT*. These correlation patterns become particularly informative when visualized graphically. The relationships between the three studied genes clearly delineate between two distinct populations, becoming apparent in pairwise gene correlations but even more pronounced when all three variables are considered together.

These findings suggest the potential for identifying biological marker combinations that could serve as clusters for distinguishing specific populations affected by psychiatric disorder. The opposing *MAOB/MB-COMT* regulation pattern, combined with *SLC6A4* downregulation, creates a unique fingerprint that could aid in OCD diagnosis, particularly in cases where clinical presentation is ambiguous or when differential diagnosis from other psychiatric conditions is challenging. Nevertheless, this remains hypothetical, as it is based on exploratory study results conducted with a relatively small sample size. The biomarker signature needs validation in larger, more diverse populations to establish sensitivity, specificity, and clinical utility thresholds.

### 3.4. Clinical Implications of Opposing Dopaminergic Enzyme Regulation and Future Directions

The opposing regulation that was observed could indeed have differential impacts across OCD symptom domains. Given that *MAOB* and *MB-COMT* regulate distinct dopaminergic pathways, with different regional brain distributions and functional roles, this enzymatic imbalance may contribute to the heterogeneous symptom presentation observed in OCD patients.

For instance, the altered dopamine metabolism in prefrontal circuits (where *COMT* is highly expressed) may be more closely linked to cognitive symptoms, such as doubt and checking behaviors. In contrast, changes in subcortical dopamine signaling (where *MAOB* plays a larger role) may relate more to repetitive motor behaviors and rituals. The simultaneous deregulation of both enzymes may elucidate the phenomenon of certain individuals exhibiting heterogeneous symptom profiles that do not conform to a single dimension.

This perspective may explain why certain patients exhibiting primarily cognitive symptoms may respond better to treatments targeting the catecholamine pathway, while individuals with predominantly motor-driven compulsions may benefit from treatments that affect MAOB-regulated metabolism. Understanding these symptom-specific enzymatic contributions could guide more personalized therapeutic strategies.

The translational potential of our findings extends beyond traditional single-target approaches, offering a pathway toward more sophisticated, personalized interventions that consider the complex interplay between serotonergic and dopaminergic systems in OCD pathophysiology.

### 3.5. Study Limitations

Given the exploratory nature of this investigation, several limitations are evident. First, PBMC samples were collected at a single time point, precluding longitudinal analysis that might better elucidate transcriptional regulation changes related to pharmacotherapy or symptomatology modifications. It is essential to recognize that the hypothesis suggesting that *MAOB* downregulation implies diminished dopamine catabolism in glial compartments. At the same time, *MB-COMT* overexpression enhances degradation at synaptic terminals; this constitutes a plausible interpretation based on the established functions of these enzymes within the brain. However, PBMCs do not possess the specialized cellular architecture of the brain, including astrocytes, radial glia, or synaptic terminals, where these enzymes primarily function in the CNS.

Moreover, the limited sample size of the present study precludes the drawing of definitive conclusions, thus hindering the stratification of data according to variables such as sex, age, therapy, and other subject characteristics. Future studies should address these limitations by evaluating all possible external factors that contribute to modulation of the epigenetic mechanism.

## 4. Materials and Methods

### 4.1. Subjects, Gene Expression, and Methylation Analysis

We included 28 OCD outpatients followed at the OCD Tertiary Outpatient Clinic at the Sacco University Hospital in Milan. Diagnoses were evaluated using a semi-structured interview based on DSM-5 criteria (SCID 5 research version, RV) [60]. For psychiatric comorbidity cases, OCD was the primary disorder, and illness severity was measured using the Yale–Brown Obsessive–Compulsive Scale [61]. Exclusion criteria included medical conditions and/or drug abuse. All patients maintained stable pharmacological treatment for at least one month, according to international guidelines [39].

The control subjects (*n* = 24) were volunteers without any psychiatric disorders, determined by nonpatient edition, with no positive family history of major psychiatric disorders among first-degree relatives [62]. The local Ethics Committee of the Sacco University Hospital approved the study protocol. Detailed demographic and clinical characteristics are reported in our previous study [63].

Nucleic acid preparation from PBMCs and gene expression analysis followed previously detailed methods [63]. In brief, PBMCs were separated by density gradient, and total RNA was isolated using modified Chomczynski and Sacchi methods [64]. After reverse transcription, mRNA abundance was assessed by real-time RT-PCR using the DNA Engine Opticon 2 Continuous Fluorescence Detection System (MJ Research, Waltham, MA 02451, USA). All data were normalized to four endogenous reference genes: *GAPDH*, *BACT*, *COX6A1*, and *RPLP0*. Relative expression was calculated using the Delta-Delta Ct (ΔΔCt) method and converted to relative expression ratios (2^−ΔΔCt^) for statistical analysis [65]. Primer sequences are provided in Appendix A.

For DNA methylation studies, we processed 500 ng genomic DNA samples through bisulfite conversion and amplified them using the Pyromark PCR Kit (Qiagen, Hilden, Germany), following the manufacturer’s protocols as previously described [66]. Primer sequences for amplification and sequencing are detailed in Appendix A (provided by Qiagen). Targeted sequences included CpG island regions upstream of *SLC6A4*, *COMT*, and *MAOB* promoter regions (see Figure 1a, Figure 2a and Figure 3a). The PCR conditions were as follows: 95 °C for 15 min, followed by 45 cycles of 94 °C for 30 s, 56 °C for 30 s, 72 °C for 30 s, and finally 72 °C for 10 min. The PCR products were verified using agarose gel electrophoresis. Pyrosequencing analysis was conducted using the PyroMark Q24 system (Qiagen).

### 4.2. Statistical Analysis

Data are expressed as mean ± standard error of the mean (SEM). Gene expression data were analyzed using nonparametric Mann–Whitney tests. The multiple *t*-tests, corrected with the Sidak–Bonferroni method, compared DNA methylation levels at individual CpG sites between groups. Spearman’s correlation analysis measured relationship strength and direction. GraphPad Prism 10 (GraphPad Software, San Diego, CA, USA) performed statistical tests and graph preparation. X-Y-Z graphs were prepared using Plot3DValues v1.31 (JAHM Software, North Reading, MA, USA).

## 5. Conclusions

Our study confirms that dysregulated monoaminergic signaling contributes to OCD development. Research indicates that OCD is a complex condition involving multiple neurotransmitter systems, particularly serotonin and dopamine. The opposing regulation of dopaminergic enzymes *MAOB* and *MB-COMT* reveals sophisticated but dysregulated compensatory mechanisms that may create spatially heterogeneous dopamine availability contributing to OCD symptomatology.

While SSRIs remain the most common and effective pharmacological option for OCD symptom relief, some patients show inadequate treatment response. Therapeutic design should consider not only the serotonergic system but also the complex interactions among different neurotransmitter systems for optimal patient-specific treatment approaches. The antagonistic dopaminergic enzyme regulation identified in this study suggests that future therapeutic strategies should target the restoration of proper enzymatic balance rather than simply enhancing or inhibiting individual pathways.

## Figures and Tables

**Figure 1 ijms-26-08811-f001:**
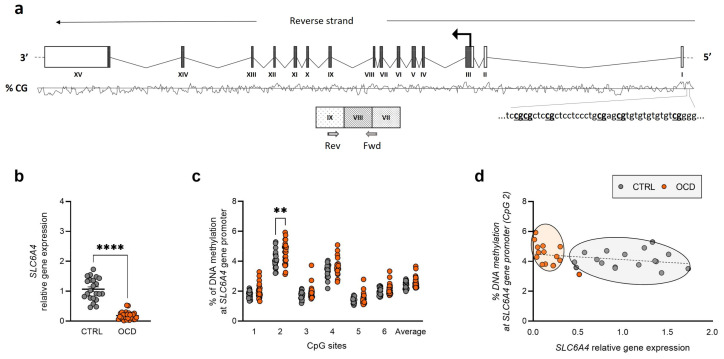
*SLC6A4* expression and regulation in human PBMCs. (**a**) Schematic representation of human *SLC6A4* gene showing translation start site (arrow), exons’ translated sequence (filled boxes), CpG island with sequences and CpG sites positions, and mRNA quantification primer locations. (**b**) *SLC6A4* relative gene expression in human PBMCs from OCD patients and healthy controls (CTRL). Scatter plots represent individual mRNA abundance calculated by the Delta-Delta Ct (ΔΔCt) method. **** *p* < 0.0001 Mann–Whitney test. (**c**) DNA methylation percentage as scatter plots for individual CpG sites and average (Ave) of 6 CpG sites. ** *p* < 0.01 multiple *t*-tests, corrected with the Sidak–Bonferroni method. (**d**) Correlation analysis between *SLC6A4* relative gene expression and DNA methylation percentage at CpG site 2.

**Figure 2 ijms-26-08811-f002:**
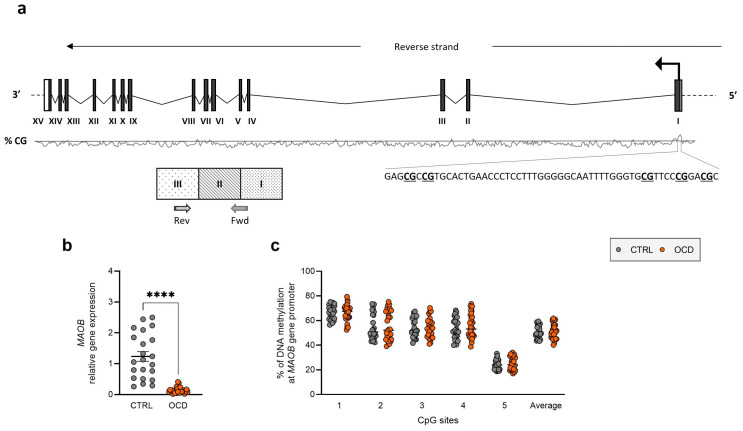
*MAOB* expression and regulation in human PBMCs. (**a**) Schematic representation of human *MAOB* gene showing translation start site (arrow), exons’ translated sequences (filled boxes), CpG island with sequences and CpG site positions, and mRNA quantification primer locations. (**b**) *MAOB* relative gene expression in human PBMCs from OCD patients and healthy controls (CTRL). Scatter plots represent individual mRNA abundance calculated by Delta-Delta Ct (ΔΔCt) method. **** *p* < 0.0001 Mann–Whitney test. (**c**) DNA methylation percentage as scatter plots for individual CpG sites and average (Ave) of 5 CpG sites.

**Figure 3 ijms-26-08811-f003:**
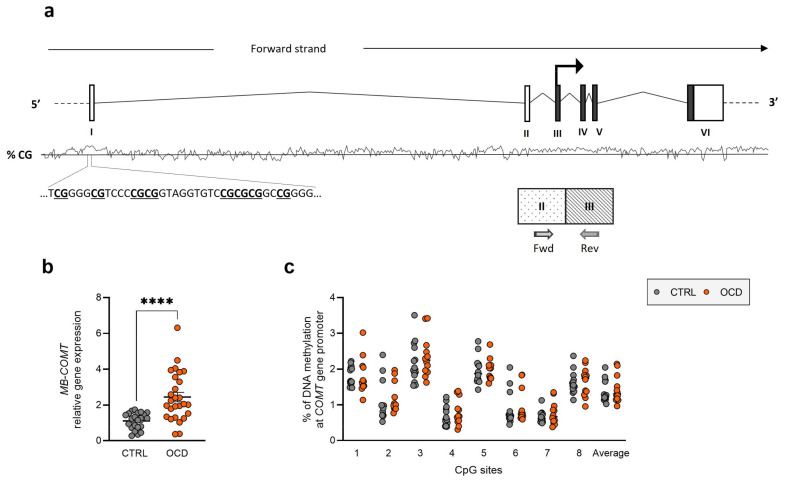
*MB-COMT* expression and regulation in human PBMCs. (**a**) Schematic representation of human *COMT* gene showing translation start site (arrow), exons’ translated sequences (filled boxes), CpG island with sequences and CpG site positions, and mRNA quantification primer locations. (**b**) *MB-COMT* relative gene expression in human PBMCs from OCD patients and healthy controls (CTRL). Scatter plots represent individual mRNA abundance calculated by Delta-Delta Ct (ΔΔCt) method. **** *p* < 0.0001 Mann–Whitney test. (**c**) DNA methylation percentage as scatter plots for individual CpG sites and average (Ave) of 8 CpG sites.

**Figure 4 ijms-26-08811-f004:**
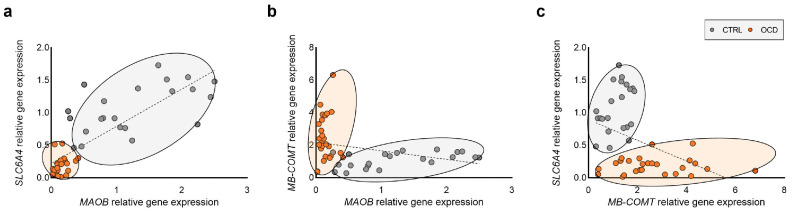
Correlation analysis between *SLC6A4* and *MAOB* (**a**), *MB-COMT* and *MAOB* (**b**), and *SLC6A4* and *MB-COMT* (**c**) in human PBMCs. Ellipses represent confidence intervals within which individuals of each group (OCD or CTRL) are clustered.

**Figure 5 ijms-26-08811-f005:**
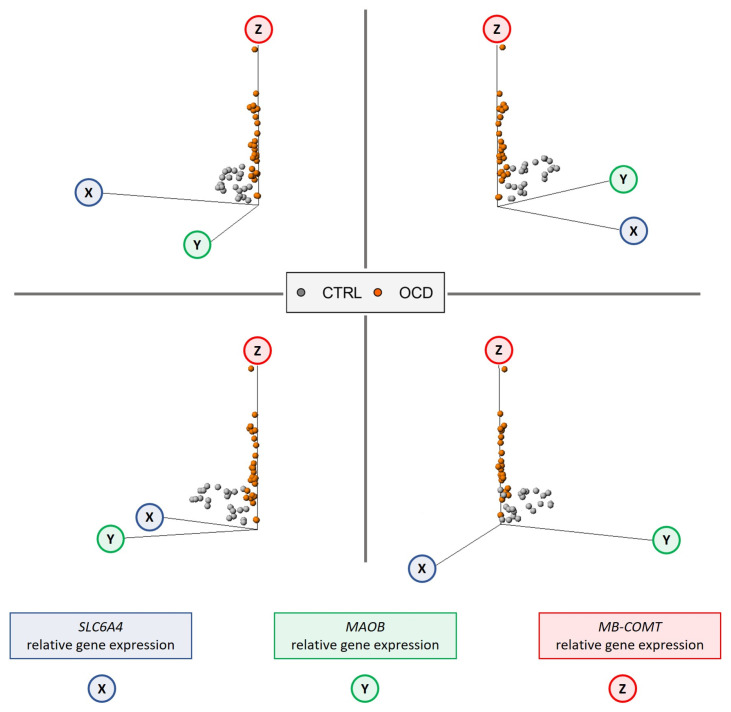
Three-component correlation analysis showing correlation between *SLC6A4* (X) and *MAOB* (Y), and *MB-COMT* (Z) gene expression.

## Data Availability

Data are available upon request.

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
