# Peer review of "Peripheral Blood Biomarkers Reveal Dysregulated Monoaminergic Pathways in Obsessive–Compulsive Disorder: A Transcriptional and Epigenetic Analysis"

_ijms, 2025, doi:10.3390/ijms26188811_

Round 1

Reviewer 1 Report

Comments and Suggestions for Authors

Review of the manuscript entitled: “Peripheral Blood Biomarkers Reveal Dysregulated Monoaminergic Pathways in Obsessive-Compulsive Disorder: A Transcriptional and Epigenetic Analysis”

General Assessment

The manuscript investigates transcriptional and epigenetic regulation of three key monoaminergic genes (SLC6A4, MAOB, and MB-COMT) in peripheral blood mononuclear cells (PBMCs) from patients with obsessive-compulsive disorder (OCD). SLC6A4 and MAOB were shown to be reduced significantly, whereas MB-COMT was increased, along with SLC6A4 promoter hypermethylation. Correlation analyses reviled interconnected expression patterns that distinguish patients from controls.

The study is well written, logically structured, and provides novel insights into potential peripheral biomarkers in OCD. It contributes significantly to the field of neuropsychiatric research by linking transcriptional and epigenetic signatures in PBMCs to OCD pathophysiology.

However, I have several concerns that authors should address before the publication of the manuscript.

Sample size – The sample size is small (28 OCD patients, 24 controls), limiting statistical power and generalizability. Stratified analyses (e.g., sex, treatment type, symptom subtypes) are too small.

Clinical heterogeneity – Even though patients were diagnosed according to DSM-5 and treated according to guidelines, heterogeneity in drug use and disease severity might have impacted gene expression profiles. Detailed subgroup analyses are not available.

Peripheral vs. central mechanisms – While PBMCs hold promise as a surrogate model, the assumption that peripheral alterations fully mirror central nervous system processes should be used cautiously. More robust validation by neuroimaging or CSF biomarkers would make conclusions definitive.

Limited epigenetic scope – Only promoter DNA methylation was assayed. Other epigenetic modifiers (e.g., histone marks, microRNAs) are likely also critical and should be assessed in future studies.

Specific Comments

Abstract: Concise and informative, but could more clearly emphasize the clinical relevance of the findings.

Results: Statistical data (effect sizes, confidence intervals) should be supplemented with p-values. Supplementary material is helpful, but a summary table of all gene expression and methylation results would be helpful.

Discussion: The section accurately describes the opposite MAOB/MB-COMT regulation, but a little extra information regarding how this imbalance may relate to specific OCD symptom dimensions would be of greater clinical utility.

Conclusion: Appropriately conservative but might better characterize translational significance, particularly for biomarker development or therapeutic use.

Recommendation

The manuscript addresses an important gap in OCD literature and presents interesting preliminary findings on transcriptional and epigenetic disturbance in PBMCs. Upon some rephrasing of limitations in discussion, provision of more statistical data, and slight elaboration on clinical significance, the study is publishable.

Recommendation: Minor Revision

Author Response

The manuscript investigates transcriptional and epigenetic regulation of three key monoaminergic genes (SLC6A4, MAOB, and MB-COMT) in peripheral blood mononuclear cells (PBMCs) from patients with obsessive-compulsive disorder (OCD). SLC6A4 and MAOB were shown to be reduced significantly, whereas MB-COMT was increased, along with SLC6A4 promoter hypermethylation. Correlation analyses reviled interconnected expression patterns that distinguish patients from controls.

The study is well written, logically structured, and provides novel insights into potential peripheral biomarkers in OCD. It contributes significantly to the field of neuropsychiatric research by linking transcriptional and epigenetic signatures in PBMCs to OCD pathophysiology.

However, I have several concerns that authors should address before the publication of the manuscript.

Comment 1: Sample size – The sample size is small (28 OCD patients, 24 controls), limiting statistical power and generalizability. Stratified analyses (e.g., sex, treatment type, symptom subtypes) are too small.

Response 1: We thank the reviewer for this important observation. We acknowledge that our sample size limits the generalizability of our findings and prevents meaningful stratification by sex, pharmacological treatment, or symptom severity. This limitation is addressed in the “Study Limitations section”.

Comment 2: Clinical heterogeneity – Even though patients were diagnosed according to DSM-5 and treated according to guidelines, heterogeneity in drug use and disease severity might have impacted gene expression profiles. Detailed subgroup analyses are not available.

Response 2: Thank you for this valuable comment. Due to our limited sample size, subgroup stratification by therapy was not feasible. As noted in our Study Limitations, this exploratory cross-sectional study would require longitudinal analysis to better understand how pharmacotherapy modulates gene regulatory mechanisms.

Comment 3: Peripheral vs. central mechanisms – While PBMCs hold promise as a surrogate model, the assumption that peripheral alterations fully mirror central nervous system processes should be used cautiously. More robust validation by neuroimaging or CSF biomarkers would make conclusions definitive.

Response 3: We thank the reviewer for the comment. We fully agree with the caution to be exercised in stating that the alterations observed in peripheral tissue completely mirror those of the CNS. Indeed, as reported in the introduction, we chose to use PBMCs to conduct these analyses considering that they share the origins and regulatory systems of neuronal tissues. Furthermore, as we have previously observed, the gene expression alterations observed in PBMCs from these subjects have also been observed in brain regions of an OCD-like preclinical model, confirming the healthy nature of this peripheral tissue. However, we would like to emphasize that the analyses performed in this peripheral tissue cannot completely replace those of the CNS, as discussed in the first paragraph. (see D’Addario et al., 2021 doi:10.2174/0929867328666211208115536, and Arosio et al., 2014 doi:10.1155/2014/169203).

Comment 4: Limited epigenetic scope – Only promoter DNA methylation was assayed. Other epigenetic modifiers (e.g., histone marks, microRNAs) are likely also critical and should be assessed in future studies.

Response 4: This limitation is recognised, and the analysis of gene regulation mechanisms will be considered in future studies, in addition to the investigation of whether protein levels reflect mRNA abundance.

Comment 5: Abstract: Concise and informative but could more clearly emphasize the clinical relevance of the findings.

Response 5: We thank the reviewer for the comment. As suggested, we have made some changes to emphasize the clinical relevance of the findings.

Comment 6: Results: Statistical data (effect sizes, confidence intervals) should be supplemented with p-values. Supplementary material is helpful, but a summary table of all gene expression and methylation results would be helpful.

Response 6: We thank the reviewer for their comment. We have included the exact CI and p-values for the gene expression analysis in Supplementary Table 1.

Comment 7: Discussion: The section accurately describes the opposite MAOB/MB-COMT regulation, but a little extra information regarding how this imbalance may relate to specific OCD symptom dimensions would be of greater clinical utility.

Response 7: We thank the reviewer for the comment. We included more details in the last paragraph of the discussion.

Comment 8: Conclusion: Appropriately conservative but might better characterize translational significance, particularly for biomarker development or therapeutic use.

Response 8: We thank the reviewer for the comment. We now improved the last part of discussion, focusing on the potential therapeutic use.

Reviewer 2 Report

Comments and Suggestions for Authors
  1. The most of the figures are not very clear, and the testing methods or calculation formulas are not given.
  2. As shown in the figure 1, figure 4 and figure 5, the clusters results really shows very significantly different among two kinds of patients and health people, I mean the clusters so clearly separate divided, that whether means that OCD is so related to the genetic factors? If not, whether the results showed the factors affect so important that could so easier to classify OCD patients? I think it’s impossible or not easier to calculate, because of OCD disease complex causes, not only contain genetic reasons, but also external influencing factors are also very critical. Genetic reason is not the main reason for OCD disease causes.
  3. Actually the ratio of OCD patients between men and women are different, but the results showed in this paper that no significant difference, why?
  4. What’s the meaning of figure 5? Give more detailed discussion for it.

Author Response

Comment 1: The most of the figures are not very clear, and the testing methods or calculation formulas are not given.

Response 1: We thank the reviewer for the comment. We represent the databased on previous published papers using similar techniques and analysis. As explained in figure captures, Figures 1a, 2a, and 3a are schematic representations of the studied genes. The graphical representation was created taking inspiration from the main gene sequence databases (see Ensembl genome browser), thus ensuring the replicability of the experiments. In the dot plots, each dot represents individual values from gene expression and DNA methylation analyses (single sites and averages). Statistical methods are detailed in Sections 2.1, 2.2, 2.3, and 4.2. Complete statistical data (group means, SEM, confidence intervals, and p-values) are provided in Supplementary Tables 1-4.

Comment 2: As shown in the figure 1, figure 4 and figure 5, the clusters results really shows very significantly different among two kinds of patients and health people, I mean the clusters so clearly separate divided, that whether means that OCD is so related to the genetic factors? If not, whether the results showed the factors affect so important that could so easier to classify OCD patients? I think it’s impossible or not easier to calculate, because of OCD disease complex causes, not only contain genetic reasons, but also external influencing factors are also very critical. Genetic reason is not the main reason for OCD disease causes.

Response 2: We thank the reviewer for this insightful observation. The clear clustering separation observed in Figures 1, 4, and 5 reflects alterations in gene expression and DNA methylation patterns rather than underlying genetic variants. We agree that OCD has complex, multifactorial causes involving both genetic predisposition and environmental factors. The distinct clustering likely indicates that despite this complexity, the dysregulation of neurotransmitter-related pathways represents a common downstream molecular signature in OCD patients. Our study focuses on these expression and methylation patterns as potential biomarkers, not as indicators of genetic causation.

Comment 3: Actually, the ratio of OCD patients between men and women are different, but the results showed in this paper that no significant difference, why?

Response 3: We thank the reviewer for the comment. Although the number of men and women differs, we did not observe substantial differences in the expression and regulation of the genes under study. On the one hand, this result allows us to propose the observed modulation of peripheral markers as potential biomarkers of OCD in general rather than of a subpopulation.

Comment 4: What’s the meaning of figure 5? Give more detailed discussion for it.

Response 4: We thank the reviewer for the comment. Figure 5 illustrates the three-dimensional interaction among the studied genes. By plotting 2^(-ΔΔCt) values representing relative mRNA levels, we demonstrate how the interaction pattern of these three genes discriminates between OCD patients and healthy controls. The direct correlation between SLC6A4 and MAOB, combined with inverse relationships between MB-COMT and both MAOB and SLC6A4, results in distinct clustering: OCD subjects’ group along the upper region of the Z-axis (MB-COMT expression), while healthy individuals cluster centrally with lower values across all three axes. This clear separation between populations reflects the coordinated dysregulation of these neurotransmitter-related genes in OCD, supporting their potential utility as peripheral biomarkers. However, we acknowledge that this molecular signature represents downstream effects of the complex, multifactorial aetiology of OCD rather than primary genetic causation.

Reviewer 3 Report

Comments and Suggestions for Authors

The present manuscript by Fabio Bellia provides an interesting investigation into the peripheral molecular alterations associated with obsessive-compulsive disorder (OCD), focusing on the expression and epigenetic regulation of three key monoaminergic genes—SLC6A4, MAOB, and MB-COMT—in peripheral blood mononuclear cells (PBMCs). The study shows distinct gene expression patterns and a possible epigenetic mechanism (hypermethylation of SLC6A4) contributing to serotonin and dopamine dysregulation in OCD. These findings are of considerable interest and may contribute to the development of objective molecular biomarkers for OCD diagnosis or therapeutic monitoring. The manuscript is generally well-structured and the experimental rationale is sound. However, several issues should be addressed to strengthen the scientific rigor and interpretability of the study.

  1. Abstract length exceeds journal limits

 The abstract currently contains approximately 293 words, which exceeds the IJMS guideline of 200 words for original articles. Please condense the abstract while preserving the key findings and interpretations.

  1. Overinterpretation of compensatory mechanisms

The hypothesis that opposing regulation of MAOB and MB-COMT represents a spatially segregated compensatory mechanism in glia vs. synapses is interesting. However, this interpretation is speculative given the use of PBMCs as the sole tissue source. The authors should more explicitly acknowledge the limitations of peripheral gene expression analysis when drawing inferences about CNS-specific processes.

  1. Paragraph unity in the Discussion

The Discussion section, though rich in content, is somewhat fragmented and lacks paragraph-level coherence in several places.

For example, lines 179–195 transition abruptly from gene expression findings to clinical pharmacotherapy without appropriate contextual framing.

Similarly, lines 211–227 present a single conceptual point—dopaminergic compensatory regulation—across multiple short and disjointed paragraphs. This content would benefit from consolidation into a single, focused paragraph.

I strongly recommend restructuring the Discussion section using a clearer paragraphing strategy based on thematic cohesion—for instance, organizing content around serotonin-related findings, dopamine-related findings, epigenetic regulation, study limitations, and clinical relevance.

  1. Clarification of Patient Cohort Characteristics

The manuscript currently lacks sufficient detail regarding the clinical characteristics of the OCD patient cohort, including age distribution, sex ratio, medication status (e.g., SSRI use), and presence of comorbid psychiatric or neurological conditions. These variables are critical for evaluating potential confounding factors that may influence gene expression or epigenetic patterns.

For example, SSRI treatment is known to affect SLC6A4 expression and methylation levels. Likewise, sex differences have been reported in COMT expression and function. Without this information, it is difficult to assess whether the observed molecular signatures are specific to OCD pathology or influenced by other variables.

I strongly recommend that the authors include a detailed table or section describing the clinical profiles of the participants, along with a discussion of how these factors were controlled for—or their possible impact on the results if they were not.

  1. Comparison with Prior Studies

The manuscript notes that the current findings contrast with previous studies, particularly regarding SLC6A4 expression in OCD patients. While this discrepancy is acknowledged, the discussion lacks a thorough explanation of the potential reasons for these differences. I recommend that the authors expand this section by critically examining factors that may account for the divergent results.

  1. English language editing.

While the overall English is readable, a professional English proofreading may help improve flow, especially in the Discussion section where some redundancy is observed.

Minor Revisions

Terminology consistency: Please standardize gene names and protein names according to HGNC and UniProt conventions (e.g., italicize gene names like SLC6A4, use all caps for human gene symbols, etc.).

Line 193: Typo: "Serotonin o plays a crucial role..." → remove extraneous "o".

Line 199: Extra semicolon after reference [43]; should be removed.

Author Response

The present manuscript by Fabio Bellia provides an interesting investigation into the peripheral molecular alterations associated with obsessive-compulsive disorder (OCD), focusing on the expression and epigenetic regulation of three key monoaminergic genes—SLC6A4, MAOB, and MB-COMT—in peripheral blood mononuclear cells (PBMCs). The study shows distinct gene expression patterns and a possible epigenetic mechanism (hypermethylation of SLC6A4) contributing to serotonin and dopamine dysregulation in OCD. These findings are of considerable interest and may contribute to the development of objective molecular biomarkers for OCD diagnosis or therapeutic monitoring. The manuscript is generally well-structured and the experimental rationale is sound. However, several issues should be addressed to strengthen the scientific rigor and interpretability of the study.

Comment 1: Abstract length exceeds journal limits

The abstract currently contains approximately 293 words, which exceeds the IJMS guideline of 200 words for original articles. Please condense the abstract while preserving the key findings and interpretations.

Response 1: We thank the reviewer for the comment. We changed the abstract as suggested.

Comment 2: Overinterpretation of compensatory mechanisms

The hypothesis that opposing regulation of MAOB and MB-COMT represents a spatially segregated compensatory mechanism in glia vs. synapses is interesting. However, this interpretation is speculative given the use of PBMCs as the sole tissue source. The authors should more explicitly acknowledge the limitations of peripheral gene expression analysis when drawing inferences about CNS-specific processes.

Response 2: We thank the reviewer for the comment. We now acknowledge the limitations linked to PBMCs use in the Study Limitation section.

Comment 3: Paragraph unity in the Discussion

The Discussion section, though rich in content, is somewhat fragmented and lacks paragraph-level coherence in several places.

For example, lines 179–195 transition abruptly from gene expression findings to clinical pharmacotherapy without appropriate contextual framing.

Similarly, lines 211–227 present a single conceptual point—dopaminergic compensatory regulation—across multiple short and disjointed paragraphs. This content would benefit from consolidation into a single, focused paragraph.

I strongly recommend restructuring the Discussion section using a clearer paragraphing strategy based on thematic cohesion—for instance, organizing content around serotonin-related findings, dopamine-related findings, epigenetic regulation, study limitations, and clinical relevance.

Response 3: We thank the reviewer for the comment. We now improved the Discussion section, restructuring with paragraphs based on thematic cohesion.

Comment 4: Clarification of Patient Cohort Characteristics

The manuscript currently lacks sufficient detail regarding the clinical characteristics of the OCD patient cohort, including age distribution, sex ratio, medication status (e.g., SSRI use), and presence of comorbid psychiatric or neurological conditions. These variables are critical for evaluating potential confounding factors that may influence gene expression or epigenetic patterns.

For example, SSRI treatment is known to affect SLC6A4 expression and methylation levels. Likewise, sex differences have been reported in COMT expression and function. Without this information, it is difficult to assess whether the observed molecular signatures are specific to OCD pathology or influenced by other variables.

I strongly recommend that the authors include a detailed table or section describing the clinical profiles of the participants, along with a discussion of how these factors were controlled for—or their possible impact on the results if they were not.

Response 4: We thank the reviewer for the comment. Although we agree that demographic and clinical information from the subjects allows for better interpretation and evaluation of the observed results, we avoided including a table containing the information of the subjects included in the study as it has already been reported in two other works by our group (see D'Addario et al., 2019 doi:10.1016/j.jpsychires.2019.04.006, D’Addario et al., 2021 doi:10.2174/0929867328666211208115536, and D'Addario et al., 2022 doi:10.1186/s13148-022-01264-0). However, if the reviewer deems it necessary to include a table with sociodemographic and clinical data, we will include it in the Supplementary Materials.

Comment 5: Comparison with Prior Studies

The manuscript notes that the current findings contrast with previous studies, particularly regarding SLC6A4 expression in OCD patients. While this discrepancy is acknowledged, the discussion lacks a thorough explanation of the potential reasons for these differences. I recommend that the authors expand this section by critically examining factors that may account for the divergent results.

Response 5: We thank the reviewer for the comment. We now included a deeper explanation hypothesis for these differences, based on methodological differences, sample heterogeneity, and comorbid conditions in the previous studies.

Comment 6: English language editing.

While the overall English is readable, a professional English proofreading may help improve flow, especially in the Discussion section where some redundancy is observed.

Response 6: We thank the reviewer for the comment. We now improved the text of the manuscript, with a specific focus on the Discussion section.

Minor Revisions

Comment 7: Terminology consistency: Please standardize gene names and protein names according to HGNC and UniProt conventions (e.g., italicize gene names like SLC6A4, use all caps for human gene symbols, etc.).

Line 193: Typo: "Serotonin o plays a crucial role..." → remove extraneous "o".

Line 199: Extra semicolon after reference [43]; should be removed.

Response 7: We thank the reviewer for the comment. We checked all the gene names and protein named according to HGNC convention and checked the presence of errors in the manuscript.

Round 2

Reviewer 2 Report

Comments and Suggestions for Authors

Fine.

Reviewer 3 Report

Comments and Suggestions for Authors

I have no concerns.